# The Use of an Electronic Painting Platform by Family Caregivers of Persons with Dementia: A Feasibility and Acceptability Study

**DOI:** 10.3390/healthcare10050870

**Published:** 2022-05-09

**Authors:** Angela Y. M. Leung, Teris Cheung, Tommy K. H. Fong, Ivy Y. Zhao, Zarina N. Kabir

**Affiliations:** 1School of Nursing, The Hong Kong Polytechnic University, Hong Kong 999077, China; teris.cheung@polyu.edu.hk (T.C.); yan-ivy.zhao@polyu.edu.hk (I.Y.Z.); 2WHO Collaborating Centre for Community Health Services (WHOCC), The Hong Kong Polytechnic University, Hong Kong 999077, China; 3Centre for Gerontological Nursing (CGN), The Hong Kong Polytechnic University, Hong Kong 999077, China; tommykhf@hku.hk; 4Department of Psychiatry, The University of Hong Kong, Hong Kong 999077, China; 5Department of Neurobiology, Care Sciences and Society, Karolinska Institute, 14183 Stockholm, Sweden; zarina.kabir@ki.se

**Keywords:** painting, mobile app, dementia, caregivers, stress, depression

## Abstract

Painting is a well-known method for alleviating stress, but it is uncertain whether family caregivers can use an electronic painting platform at home for this purpose. Aim. The aim in this study was to assess the feasibility and acceptability of having family caregivers of persons with dementia (FCPWD) draw electronic paintings using a mobile app, and to assess the preliminary effect of the intervention on their well-being. Methods. This was a two-phase feasibility and acceptability study, with qualitative interviews conducted in Phase 1 and qualitative interviews and a quantitative survey conducted in Phase 2. Caregiving burden, depressive symptoms, self-rated health, and social support were measured before and after the intervention. Participants were asked to draw electronic paintings at any time they liked, and to share the paintings with friends or relatives if they wished. Result. The recruitment rate in Phase 2 was 87.5% (28 out of 32), with 78.6% participants (22 out 28) completing all activities in 8 weeks. The FCPWD regarded the e-painting app as an appropriate channel for expressing their emotions. They found the layout of the app to be easy to use and were satisfied with it. A total of 116 pictures were produced. Log-in frequency was significantly correlated with the sharing of paintings with friends or relatives (r = 0.72, *p* < 0.001). Conclusion. FCPWD considered the e-painting mobile app to be a feasible and acceptable technology-based psychosocial platform. A further investigation with a larger sample in a full-scale randomized controlled trial is warranted.

## 1. Introduction

Dementia affects not only persons with the condition but also their family caregivers. It is estimated that by 2050, the number of people around the globe suffering from dementia will rise to approximately 152 million [1]. Undoubtedly, family caregivers of persons with dementia (FCPWD) will continue to experience different challenges in carrying out their caregiving tasks.

### 1.1. Family Caregivers of Persons with Dementia Experience Distress

FCPWD have been shown to be at a higher risk of anxiety, depression, and distress compared to age-matched controls [2]. Such mental health conditions mainly originate from the demands of performing day-to-day caregiving tasks (such as bathing and assisting with toileting) and from the burden of care, which is exacerbated by the behavior of the care recipients. FCPWD often feel anxious about the wandering behavior of care recipients. FCPWD become depressed or even severely distressed when handling changes in the behavior of care recipients in public. Evidence shows that FCPWD who deal with prevalent behavioral problems have a significantly higher chance of developing depression than their counterparts (OR = 5.2, *p* < 0.001) [3]. Caregivers who take care of people with memory disorders are 8.4 times more likely to experience distress than those who care for persons without memory disorders [3]. As dementia progresses to an advanced stage, the degree of anxiety and depression experienced by FCPWD increases [3]. In China, about one quarter of FCPWD suffer from anxiety (29%) and depression (21%) [4]. A recent meta-analytic review confirmed that caregivers experience worse mental health than non-caregivers [5]. FCPWD in Hong Kong have reported experiencing a high level of distress due to caregiving [6].

Caregiver stress has been defined as ‘an unequal distribution of responsibilities on the caregiver as a result of caring for someone with a prolonged impairment’ [7]. Providing care to persons with dementia imposes a tremendous amount of stress on caregivers because people with dementia often exhibit behaviors that other people find confrontational [8]. The dependence of patients in the activities of daily life, and the severity and frequency of their behavioral symptoms, have been found to be significantly associated with stress-related depressive symptoms in caregivers [9]. Another major source of stress is caregivers’ subjective feelings of being overloaded [10].

Interventions have been carried out to reduce the burden on and distress experienced by caregivers. Most of the interventions involved face-to-face weekly sessions, but this design may have been too demanding for caregivers because many of them were reluctant to leave the care recipients to attend these face-to-face activities. Existing interventions do not meet the needs of all caregivers. Art therapy, particularly painting, seems to be a promising alternative approach, with one review concluding that a brief period of painting significantly reduced distress and depression [11]. With advancements in technology, a number of online interventions have been developed to support caregivers [12], and these give them the flexibility to participate. Digital art therapy, using technological devices to create images, has been considered a highly promising intervention for boosting mental well-being [13]. An exploratory study concluded that painting in mobile apps (e-painting) can be beneficial to persons with anxiety issues because many people are familiar with digital media [13]. E-painting gives persons with distress a means to understand themselves, distracts them from their troubles, and allows them to express themselves and relax [13]. An e-painting app has been shown to be acceptable for use by persons with post-traumatic stress disorders and feasible in terms of managing stress symptoms [14]. However, whether e-painting has a similar effect on other populations, particularly vulnerable caregivers, is unclear.

### 1.2. Use of Mobile Applications in Stress Management

Mobile applications (apps) have been developed for stress management. Persons suffering from post-traumatic stress disorders were selected as the target population to assess the feasibility and acceptability of using such apps to manage acute stress or stress symptoms [14]. Users considered such apps to be an ‘acceptable’, helpful’ and ‘effective’ self-management tool for persons experiencing stress [14]. A mobile app-based intervention using mindfulness-based cognitive-behavioral therapy significantly reduced the stress and depressive symptoms of caregivers who were providing care to family or friends with a physical or mental disability [15].

### 1.3. Painting in Stress Management

Evidence has shown that art therapy reduces stress symptoms among pediatric patients [16], working adults [17], and older adults [18]. Using functional magnetic resonance imaging (fMRI), Bolwerk and team [19] found that those who participated in visual art production (painting) showed greater spatial improvement in the functional connectivity of their posterior cingulated cortex to their frontal and parietal cortices than those who cognitively evaluated artwork in a museum. This implied that painting, but not the viewing of artwork, had an effect on psychological resilience in adulthood [19]. A review [11] summarized these results, indicating that a brief period of painting significantly reduced anxiety [20,21], distress [22], and depression [23] and also increased mental alertness and sociability [24].

Traditional artwork includes the viewing of art (in a gallery), the making of art, or both. All previous studies [25,26,27] about ‘artwork and caregiving’ included both care recipients (e.g., persons with dementia) and caregivers, but none was specially designed for caregivers of people with dementia. Twelve caregivers, together with 12 persons with dementia, participated in the 8-week art-viewing and art-making intervention. No significant change in carer burden was observed, although there was a downward trend [26]. However, in the qualitative interviews, these caregivers indicated that they valued the time spent in making art as an opportunity to take a break and to connect with others, thereby improving their social network [25,26]. Similarly, in another study on traditional artwork for dementia care, 30 caregivers said that they appreciated the building of a relationship with the care recipient through the art-making activities [27]. To date, the link between online art and dementia care has not been well established—some modules have recently been set up in the USA, but evaluations have yet to be carried out [27]. One study, using artwork to support caregivers of children who died from cancer, showed that the caregivers experienced fewer symptoms of prolonged grief after engaging in an art intervention [28].

The creation of digital art reportedly offered people another option to access art and to express their emotions [29]. However, there were barriers to engaging in digital art compared with traditional artwork, such as a lack of digital inclusion [29]. Nonetheless, evidence on the creating of digital artwork by the family caregivers of people with significant chronic illnesses, including dementia, is limited. Embedded in the knowledge of painting and its possible impact on the psychological well-being (anxiety, stress, depression) and sociability of caregivers, this study aims to (1) examine the feasibility of using an electronic platform (a mobile app) to support FCPWD to draw and share paintings; (2) examine the acceptability to FCPWD of using this e-painting app; and (3) assess the preliminary effect of this intervention on the psychosocial well-being of caregivers.

## 2. Materials and Methods

### 2.1. Trial Registration

The trial has been registered in the trial register ClinicalTrials.gov (https://clinicaltrials.gov/ct2/show/NCT03850613 accessed on 14 March 2022). 

### 2.2. Design

This was a two-phase study with a mixed-methods design (including a pre- and post-quantitative survey and post-intervention semi-structured interviews). Phase 1 was the development of the e-painting app while Phase 2 involved testing the feasibility and acceptability of the app. To develop an app that was user-friendly and satisfactory to FCPWD, interviews were held with a group of FCPWD in Phase 1 before the app was developed (version 1.0). The app was then revised according to the preferences and recommendations of the FCPWD, and became version 1.1. In Phase 2, FCPWD were invited to use the e-painting mobile app (version 1.1) for 8 weeks. They were invited to complete a survey before and after using the e-painting app and to participate in a semi-structured focus group interview after using the app. Since there has been no previous study on electronic painting, reference was made to two previous studies on traditional painting for mental health [17,30] and to the usual duration of community-based interventions for caregivers in Hong Kong. The duration of the two interventions in previous studies ranged from 4 to 10 weeks [17,30]. Therefore, we chose 8 weeks as the duration for the intervention in the current study. Since some participants could not join the scheduled focus group interviews, they were interviewed one-on-one either in a face-to-face mode or via telephone. Open-ended questions were used to elicit data from the FCPWD in the focus group interviews. The qualitative data that emerged from the interviews were triangulated and corroborated with the quantitative survey data [31]. A broader and more in-depth understanding of the views of FCPWDs on the use of mobile app was sought.

### 2.3. Sampling and Sample Size

Convenience sampling was used in this study. Twenty-five samples in a feasibility and acceptability study were considered adequate [32]. Assuming an attrition rate of 20%, the total sample size of the proposed study was 32.

### 2.4. Participants and Inclusion and Exclusion Criteria

To be included in this study, participants had to be aged 18 or above, have a family member providing direct care to persons with dementia, and be able to understand/speak/read Chinese (either Cantonese or Putonghua). Family caregivers who were hospitalized, residing in residential care homes or medically unfit, or who had manual dexterity or vision issues were excluded from this study.

### 2.5. Recruitment and Consent

Four local non-governmental organizations assisted in identifying potential participants for this study. Flyers were posted in the communal areas of community centers. Potential participants who met the inclusion criteria were contacted by a research assistant by telephone. A briefing session, which was in a face-to-face format, was scheduled for the potential participants to explain the aim and objectives of this study and data collection procedures. Before the end of the briefing session, those who agreed to join the study downloaded the e-painting mobile app on their mobile phones and engaged in 20 min of hands-on practice. Five core functions in the app (1: picture sharing; 2: painting; 3: chatroom; 4: announcement; 5: self-assessment) were introduced in this practice session. The project team answered the participants’ queries, if any. Opportunities were given to each participant to express any concerns or anticipated difficulties before the commencement of the 8-week intervention.

### 2.6. Data Collection

The participants were encouraged to use the e-painting app to draw at least two paintings every week at their leisure on a portable device (either a mobile phone or a tablet) with a finger or a stylus (if their mobile phone included a stylus). They could use different types of brushes (a watercolor brush, oil painting brush, acrylic brush, etc.) and a variety of color fillings in the app to assist in creating their paintings. There was no restriction on the number of paintings they produced during the whole period (8 weeks). The participants were informed that their paintings could be uploaded to a secure server as well as be contained in their mobile phones. They were welcome to share their paintings with friends or relatives. On a weekly basis, the administrator announced a theme for drawing the paintings in the ‘Announcement’ feature (Appendix A). Other than drawing the paintings, the participants were asked to undergo a self-assessment using the app at Week 1 and Week 8. The data collection period was from February to June 2018.

#### Recruitment Rate, Completion Rate, and Retention Rate

Figure 1 shows the flowchart of the recruitment process and measurements at different times. A total of 50 FCPWD (22 in Phase 1 and 28 in Phase 2) were involved in the whole study. Twenty-two of them (7 males) participated in the focus group interviews in Phase 1 (development of the app), indicating their preferences with regard to the features of the e-painting mobile app and their training needs. Among the 32 FCPWD who were approached, 28 participated in the 8-week intervention. The recruitment rate in Phase 2 was 87.5%. The completion rate for all activities in 8 weeks was 78.6% (22 out of 28). After the intervention, 50% of the participants (n = 14) took part in the post-intervention interviews, either in a focus group, on a one-to-one basis, or in a telephone interview.

### 2.7. Ethical Issues

Ethical approval was granted by the Human Subjects Research Ethics Subcommittee of the Hong Kong Polytechnic University (Reference #: HSEARS20180126008). Participants were informed of the purpose, procedures, and contents of the study. Each participant was allocated a confidential unique study ID for follow-up data collection and assured of confidentiality and anonymity. Written informed consent was sought from all of the participants prior to the use of the app as well as before the focus group interviews were conducted. All study documents were stored in a locked filing cabinet at the study university. Only the members of the project team have access to the consent forms.

### 2.8. Quantitative Survey

#### 2.8.1. Background Information

Sociodemographic information was solicited, including data on gender, age, marital status, educational attainment, employment status, occupation, monthly household income, and time for providing care to persons with dementia.

#### 2.8.2. Psychosocial Assessments

##### Caregivers’ Burden

Caregivers’ burden was measured using the 12-item short Chinese version of the Zarit Burden Interview (CZBI-Short) [33]. The CZBI-Short is a valid and reliable instrument with a Cronbach’s alpha of 0.84. The respondents were asked to rate their level of burden associated with caring for the person with dementia on a 4-point Likert scale (0: ‘Never’; 1: ‘Rarely’; 2: ‘Sometimes’; 3: ‘Quite frequently’; 4: ‘Nearly always’). The total score ranges from 0 to 48 (0–10: no to mild burden; 10-20: mild to moderate burden; >20: high burden).

##### Self-Rated Health (SRH)

The respondents were asked to give an overall rating of their health status in the past three months on a single-item 5-point Likert Scale ranging from 1 to 5 (1: excellent; 5: poor) [34]. The higher the score, the poorer the SRH.

##### Depressive Symptoms

Depressive symptoms were measured using the Patient Health Questionnaire-9 (PHQ-9)], a 9-item self-administrative questionnaire that asks the respondents to indicate the frequency of some typical events/feelings (e.g., little interest/pleasure in doing things, feeling down, depressive, hopeless, feeling tired/have little energy) in the past two weeks on a 4-point Likert scale from 0 to 3 (0: not at all; 3: nearly every day) [35]. Scores in the PHQ-9 range from 0–27. The higher the score, the more severe the depressive symptoms [35]. A score of 5, 10, 15, and 20 represents the cut-off points of ‘mild’, ‘moderate’, ‘moderately severe’, and ‘severe’ depressive symptoms, respectively [35]. The PHQ-9 demonstrated good validity, reliability, internal consistency (Cronbach’s alpha = 0.802), and test-retest reliability (ICC = 0.988) among the general population in Hong Kong [35].

##### Instrumental and Emotional Social Support

Instrumental and emotional social support was measured using the 8-item Modified Medical Outcome Study Social Support Survey (mMOS-SS) [36]. The internal reliability and consistency, and construct and discriminant validity of the mMOS-SS were found to be excellent [36]. The mMOS-SS ranges from 1 to 5 (1: Never; 5: always), with higher scores indicating more instrumental and emotional social support [36].

### 2.9. Focus Group Interviews

Before the development of the app, six focus group interviews were conducted. After the implementation of the intervention, five focus group interviews, five individual interviews, and four phone interviews were conducted. Each interview was moderated by two researchers (A.L. and T.F.). All face-to-face interviews were held in community elderly centers. The former researcher led the discussion by posting the interview guiding questions while the latter researcher took notes during the interviews. Audio-recordings were made, and verbatim transcriptions were made and stored in the server. Each group consisted of 3–4 participants. Each focus group interview lasted for 45 min to one hour.

Three guiding questions were used in the pre-intervention interviews:What functions do you prefer in the e-painting app?What kind of training do you need before using the e-painting app?What kinds of follow-up actions do you prefer after using the e-painting app?

(e.g., painting analysis; professional counselling and support).

Four guiding questions were used in the post-intervention interviews:What do you think about this e-painting mobile app?What do you like or dislike most about this app?What features could be added to this app?Do you enjoy using the e-painting app? And in what way?

### 2.10. Statistical Analysis

A statistical analysis was performed using SPSS version 26.0 for the Windows platform (SPSS Inc.; Chicago, IL, USA). A descriptive analysis was used to report the sociodemographic characteristics of the participants. Descriptive data were presented in terms of frequency (n), percentage (%), and standard deviation (SD). A *p*-value of <0.05 was considered statistically significant with 95% confidence intervals (CI). Paired sample *t*-tests were used to compare the mean score before and after the intervention.

### 2.11. Qualitative Analysis

Qualitative data from all interviews were audio-taped, transcribed verbatim into Chinese and back-translated to English by a trained researcher (T.L). Coding was used to identify common patterns, associations, and relationships from the participants’ narratives. The coding was conducted by two independent bilingual researchers (A.L. & T.C.). Both hold a PhD in the Social Sciences, with one being an expert in dementia research while the other is a mental health specialist and educator. A code book was used as a guide and reference for the coding system. A content analysis was used to identify themes and sub-themes, which were re-contextualized to provide coherent constructs so as to examine the relationship between ideas [37]. Rigor was established through credibility, plausibility, and transferability [37].

## 3. Results

The results were triangulated from the qualitative and quantitative data and are presented below.

### 3.1. Phase 1 Result: Preferences with Regard to Features and Expected Training

The participants generally liked the idea of having an electronic platform for drawing paintings. Some participants suggested that ‘background music’ be added and ‘a wide range of color’ be used for filling in the paintings. They believed that these features could help to alleviate their stress and improve their mood.


*“Does the app have music, like background music? … Yeah, that’s my way of relaxing, music … is one of the methods.”*
(Participant 1, February 1, L557; 561)


*“For example, you can have colour filling, or some demonstrations for app users, then you can follow and imitate the drawing.”*
(Participant 2, February 1, L552–553)

A participant suggested that a mood assessment function be included in the app so that self-assessments can be done.


*“Yeah, (a mood assessment) could tell which level you are. You can have some suggestions (from the assessment results) such as what things you can do next, whether you can relax a bit, find someone to talk with, etc.”*
(Participant 28, April 20, L178–180)

The participants suggested that having some hands-on practice before using the app at home would be beneficial. They preferred to have some guidance on how to download the app, how to use different features, and how to upload or share the paintings. They would like to have some professionals analyze their paintings and provide counselling and support.

### 3.2. Phase 2 Result: Demographic Information

Table 1 shows the sociodemographic characteristics of the participants. The majority of the participants were female (71.4%), married (92.9%), and either retired or unemployed (78.6%). The mean age was 59.71 (SD 9.88). About one-third had obtained a Bachelor’s degree or above (35.7%) and half had attained a secondary school education. Two-fifths (39.3%) had a monthly income of HK$12,001 (USD 1535.99) or more. Two-fifths of the participants (39.2%) had been caring for their family member with dementia for 5 years or longer.

### 3.3. Phase 2 Result: Feasibility of the Use of the Electronic Painting Platform

Table 2 shows the frequency of logins and picture sharing. During the 8-week intervention, the majority (n = 18, 64.3%) of participants logged in 1 to 8 times, while a small number (n = 2, 7.1%) of people were identified as high-frequency users (one logged in 20 times and the other 23 times). The average number of logins was 6.89 (SD 6.53). It was quite common to share paintings. Most of the participants (n = 14, 71.4%) shared their paintings with friends/relatives while 15% shared more than 9 paintings during the intervention period. The average number of pictures shared was 3.75 (SD 5.29). A correlation analysis showed that the frequency of logins was highly correlated to the sharing of paintings (r = 0.717, *p* < 0.001).

Figure 2 showed the frequency of logins and sharing. The login intervals were counted every 24 h. The app usage of each participant was counted throughout the 8-week intervention. Login frequency and picture sharing was highest in Week 1 (45 login sessions and 54 pictures shared). However, both the frequency of logins and sharing of paintings dropped after Week 2 and reached a low point in Week 6. There was a sharp increase in login frequency in Week 7, and usage again dropped progressively by the end of Week 8. A total of 116 pictures were produced (Figure A1).

Although the electronic painting platform was scheduled to be used for 8 weeks, the platform continued to operate after that period. We observed use of the app by some participants from Week 9 to Week 16. Picture sharing also dropped progressively from Week 2 onwards, with the lowest point in Weeks 6, 11, 13, and 14, but a slight increase in in Week 8. Week 8 was officially the last week for the use of the e-painting app. Nevertheless, it was encouraging to note that some participants continued to use the app beyond the test period.

Figure 3 shows the time when the participants used the e-painting app. Five peak hours were noted: 1 am, 10 am, 3 pm, 5 pm, and 10 pm.

### 3.4. Phase 2 Result: Acceptance of this E-Painting App by the FCPWD

A quarter (*n* = 7, 25%) of the participants indicated their satisfaction with the e-painting app, and about half (53.6%) rated the app as ‘fair’. None considered the app to be ‘unsatisfactory’. Correlation analyses showed that none of the demographic characteristics was correlated with the use of the e-painting app and the action of sharing the paintings.

### 3.5. Phase 2 Result: Preliminary Efficacy of the Intervention on Psychosocial Well-Being

Table 3 reports the changes in the psychosocial well-being of the participants after the intervention. There was a significant increase in burden after the intervention (mean difference = 2.86, *t* = 3.10, *p* = 0.004). Nonetheless, there were no significant changes in self-rated health, depressive symptoms, and social support after the use of the app. This intervention had a small-to-moderate effect on caregiver burden (Cohen’s D = 0.41).

### 3.6. Phase 2 Result: Qualitative Results

Four themes were identified from the interviews, namely satisfaction and enjoyment in the use of the e-painting app; the app as a channel to ventilate emotions; the app makes me feel connected, and combatting the challenges due to caregiving.


**Theme** **1.**
*Satisfaction and enjoyment in the use of the e-painting app.*



In the focus group interviews, the FCPWD expressed their satisfaction with the app and regarded the e-painting app as ‘fun’ and ‘enjoyable’.



*“Enjoy! It [the e-painting app] is very easy to use.”*
(Participant 36, L144–147)

*“This is fun … this one [the e-painting app] is fun.”*
(Participant 5, L92–95)

*“I looked at what the others drew … sometimes, when I looked at those paintings, I found it was very interesting … it was fun.”*
(Participant 36, L161–164)


The app was also considered ‘convenient to use’ and ‘user-friendly’. There was no pressure in painting because the participants could remove the paintings if they did not like them.



*“I want to draw, I want to draw well, but I cannot draw well. It is okay not to draw well, you can point to the ‘white’ and then they will vanish. See? This is really good.”*
(Participant 5, L60–63)

*“But it is good to use the mobile phone to draw the paintings, that is, no matter how you do this … you could still draw the paintings … it is convenient [Another participant agreed and said—convenient!], the advantage here is its convenience.”*
(Participant 29), L21–252)
**Theme** **2.**
*The app as a channel to ventilate emotions.*



The FCPWD were asked about the value of this e-painting app. They regarded the app as a channel to ventilate their negative emotions. When the FCPWD were upset, they used the app to work out the paintings—this seemed to stabilize their emotions.



*“Whenever I feel annoyed, I used to throw objects or tear things apart … but now, err. Painting can soothe my bad mood…. I see it as a way to let go of bad feelings”*
(Participant 5, L55–59)

*“Yes … yes … this is a pressure releasing tool.’*
(Participant 5, L60–63)

*“Whenever I paint, I feel like I’m brushing away all the negative events.”*
(Participant 19, L282–284)

*“I became happy when I drew the paintings, that is, say … when you are drawing for two hours, you forget everything in these two hours … you do not remember the other things. You are in the paintings.”*
(Participant 29, L336–337)

*“At the beginning, I drew more, then recently I drew few paintings … my emotions have become stable these days … ah …. Those paintings … make a person more positive, and make the emotions better.”*
(Participant 29, L30–32)
**Theme** **3.**
*The app makes me feel connected.*



Many of the participants stated that they had limited or no support from other family members or their social network. Therefore, they felt that the e-painting intervention could help them and make them feel connected. They communicated in the Chatbox and shared the paintings. They knew other caregivers were in the group, watching the paintings and appraising the artwork of others. They felt connected to someone who had similar experiences due to caregiving.



*“I really find e-painting helpful.”*
(Participant 19, L270–272)

*“I wanna talk to someone but when no one is there for me … I will then try e-painting”*
(Participant 7, L621)

*“[T]hen I feel … happier. No matter whether I write this correctly or not, let people hear what I say … then someone will respond to me, someone will read my words.’ ‘Anyway … there is some communication here.”*
(Participants 25 and 26, L93–95)

*“Because of the group [in the app], whatever you put into the chat, the other caregivers can read it. That is, you know, you have already communicated with others.”*
(Participant 26, L110–111)

*“It is better to draw the paintings by yourself, you can look at others’ paintings, but you should participate … this is better, then you interact with others. … that is, we are interacting, like this … in fact, we are the same group of people … that is, we are facing similar challenges.”*
(Participant 29, L44–49, 52)
**Theme** **4.**
*Combatting the challenges due to caregiving.*



They then started to talk about the challenges they faced due to caregiving. They also found it hard to share their feelings with other people because of stigmatization:


*“[S]tigma … I don’t have the guts to tell others….’*
(Participant 3, L383–386 & L397–400)

Because of the stress due to caregiving, they experienced both physical and psychological stress.


*“It is very harsh…. [I am] physically, psychologically drained.”*
(Participant 8, L177–179)


*“Insomnia … I’m losing my appetite and didn’t sleep well … err … not sure if I’m depressive coz I have lost interest in everything.”*
(Participant 1, L378–381).


*“Istay at home almost 24 h/day and don’t go out…. I am scared of being alone at home coz … I feel very unhappy.”*
(Participant 17, L34–37)

Despite all the challenges, the FCPWD found ways to combat the challenges. One strategy was to talk to others, check out the experiences of other caregivers, and get support from their peers.


*“Do you know (we) caregivers like to get someone to talk?”*
 (Participant 5, L99–101)


*“Yes, why do we like to be together? In fact, we like something simple. Caregivers only need to ventilate, or listen to others’ experiences; this is for our reference. If I know that this caregiver is in a low mood, I will comfort him/her. Err … this is very important. No one supports us!”*
(Participant 5, L113–117)

Although two participants disagreed about the benefits of participating in e-painting, the FCPWD liked the ‘announcement’ function of the e-painting app because they wanted to get some information about caregiving skills.


*“E-painting does not seem to help alleviate my stress.”*
(Participant 7, May 20, L173–175)


*“I don’t like painting … it doesn’t help.”*
(Participant 8, May 20, L354–357)


*“If you ask me, I like the ‘announcements’—announcements of news. Because, in the group, everyone is reading the messages from this … that is, we get some advice/directions.”*
(Participant 25, L84–85)


*“I think the announcements … should be … about showing us how to take care of the illness [dementia], that is, the sick [person with dementia], to recognize the sick person. Because in many occasions, their emotions have … significantly changed.”*
(Participant 13, L184–186)

## 4. Discussion

This study was the first to explore the feasibility of using an electronic platform (a mobile app) for FCPWD to paint during their leisure time as a stress alleviation strategy. It has always been a challenge to involve FCPWD in center-based caregiver support activities/interventions because many do not like to leave the persons with dementia at home by themselves while they participate in the activities held in community centers [38]. This 8-week e-painting intervention was shown to be feasible and acceptable to the FCPWD. The high recruitment rate indicated that these caregivers welcomed the use of technology as a means of delivering an intervention. This implies the possible future development of a caregiver support intervention offered through an electronic platform.

Another strength of this study was the co-design approach in Phase 1. The involvement of stakeholders (FCPWD) in the developmental stage of the mobile app enhanced acceptance of the app by the FCPWD. Some features, for example, the mood assessment function and the availability of wide range of colors, were proposed by the FCPWD. The project team considered these suggestions and incorporated these features in the second version of the app. We also added background music to the app, which has a similar effect to that of music therapy of providing a relaxing environment for the participants to draw paintings and reduce their stress. It is crucial to adopt a co-design approach when developing mobile health tools to support laymen in health promotions and health education [39].

The use of this electronic platform (the e-painting app) for drawing and sharing paintings had no significant relationship with demographic characteristics. This may imply that people of different ages, genders, marital status, levels of education, employment status, and income levels would be inclined to use this app in a similar way. This is a good indicator, showing that this app can be used by anyone.

Painting is the expression of ideas and emotions from the inner psychological self of individuals [40]. Although no significant improvement in depressive symptoms was discerned in the quantitative analysis, the participants mentioned that they were satisfied with this intervention and asserted that the e-painting app was a ‘channel to ventilate their negative emotions’. Some participants even wrote a few words on the paintings to express their feelings and shared it with other participants. Many found this app to be acceptable, ‘easy-to-use’, and ‘enjoyable’. The participants treated this app as a method to release their emotions. Expressing emotions could only be done when the caregivers were free from daily caregiving duties. Reviewing the peak hours for using the e-painting app, we noted three periods (10 pm to 1 am; 9 to 10 am; 3 pm to 5 pm), when FCPWD were comparatively relaxed and able to work on their paintings and express their emotions. Some carers used the app between 11 pm and 1 am, which might be due to the lifestyle in a metropolitan city or to their caregiving responsibilities. Knowing these time periods may be helpful for professionals, who could use these times to communicate with the FCPWD, providing psychological support to them if necessary.

Support was extended to the participants when they used the e-painting app. Rather than using traditional painting tools, the participants used their mobile phones to draw pictures and upload them onto the cloud. Since e-painting required the participants to master some level of digital technology, we thought that repeated setbacks might cause the participants to lose interest and deter them from continuing to use the app [41]. Therefore, we provided briefing sessions and hands-on practice to support them in using this app. Continuous support and encouragement was also provided. During this 8-week intervention, the project team engaged the participants by sending them ‘a theme to draw’ and a warm reminder note to use this app (Table A1, Appendix A). As all of the participants shared the same target, i.e., to draw paintings on this theme, the majority logged in and shared their paintings at least one to two times per week.

One finding worth noting was the significant increase in the caregiving burden after the 8-week intervention. Since this study was a one-arm pre-and-post study, the change in the caregiving burden might not have been the sole effect of the intervention. The underlying reasons for the increased caregiving burden upon completion of the 8-week intervention were unclear. Unfortunately, there was no control group in this study. We could not observe the pattern of the caregiving burden in the control group and compare it with the intervention group. The current increase in caregiving burden (the change in the CZBI mean score) was statistically significant but not clinically significant. A further investigation with a larger sample size and a control group is needed. Providing care to persons with dementia is very demanding. The use of the e-painting app did not seem to help to reduce the caregiving burden. As indicated in the focus group interviews, some caregivers appreciated the experience of sharing among caregivers in Chatbox, yet this did not seem to be enough to lessen their burden. More data are needed to understand the underlying reasons for an increase in burden over time. A possible strategy for reducing the caregiving burden is to provide information about caregiving skills by professionals in the Announcement or Chatbox function of this app.

The current findings did not provide strong evidence of any improvement in self-rated health and social support, or of a reduction in depressive symptoms after the use of the 8-week e-painting intervention. However, a digital art intervention was documented to lead to a significant reduction in caregiver burden among FCPWD [42]. It is worth noting the changes in these variables after the intervention. Since the effects on self-rated health, social support, and depressive symptoms were very small, a large sample size would be needed to capture significant changes after the intervention.

Recruitment of the FCPWD in this study was supported by four non-governmental organizations, and the sample size was small. This reflected the challenges of recruiting family caregivers in an interventional study. As this intervention was in a digital format, some caregivers hesitated to join this study because of perceived limitations in terms of digital literacy, digital skills, and the possession of personal mobile devices [43]. This implies that there is a need to provide support to caregivers, including setting up a social support system and providing proper training in digital competence and digital infrastructure when digital interventions are used to support caregivers [43].

As this e-painting app was found to be feasible and acceptable in real life settings, a large-scale trial involving the collecting of longitudinal data on caregiver burden and other psychosocial health outcomes (such as resilience and a sense of coherence) is warranted after some adjustments are made to the app. The other possible step in investigation involves interpreting the paintings, in consultation with the caregivers, in relation to the emotions they felt when they drew these painting. Professionals could further investigate the psychological status of these caregivers and provide counselling and support, whenever necessary.

## 5. Limitations

First, the sample of this study was restricted to FCPWD. The findings cannot be applied to other populations such as people with complex chronic medical conditions who also need interventions to support their psychosocial health. Second, this app cannot be used by people who have limited access to Wi-Fi or electricity. This implies that some caregivers would inevitably be excluded from using this intervention. Third, the sample size was small, although the samples were from multiple centers. Fourth, there was no comparison group in this study. The actual effect of the intervention on the outcomes could not be clearly shown. Single-arm trials are often used, but such a design remains controversial because of the difficulty of distinguishing the effect of the intervention from other factors. Such a decline in a person’s health condition increases the caregiving burden on family caregivers. Therefore, it is not surprising that, in this study, the family caregivers experienced a greater caregiving burden over time (8 weeks is a period during which the condition of a person with dementia could change). In a future study, a comparison group (and a larger sample size) should be included so as distinguish the sole effect of the intervention on the caregiving burden and any other outcomes. Notwithstanding these limitations, the findings of this study show the feasibility and acceptability of this e-painting app. This is encouraging evidence to support future studies in health technology.

## 6. Conclusions

This study shows the feasibility and acceptability of an e-painting app among family caregivers of persons with dementia. Although there were no significant changes in their depressive symptoms, self-rated health, and perceived social support, the caregivers indicated their satisfaction and enjoyment in the use of this e-painting app and considered it to be a channel to express their emotions and connect with other caregivers. This 8-week intervention aroused the interest of caregivers in terms of using technology for self-care. This study demonstrated a novel psychosocial support tool combining digital technology, paintings, and music for family caregivers of persons with dementia. More evidence is needed on the potential to test the efficacy of this app on the psychosocial well-being of caregivers. The findings of this study suggest possible adjustments to consider in order to make the app more user-friendly before a larger scale trial is implemented.

## Figures and Tables

**Figure 1 healthcare-10-00870-f001:**
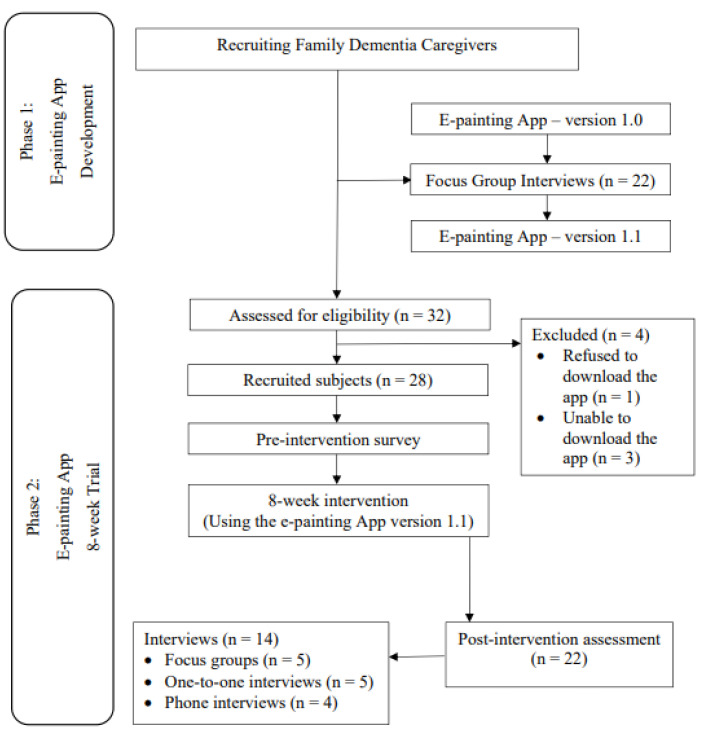
Flowchart Diagram.

**Figure 2 healthcare-10-00870-f002:**
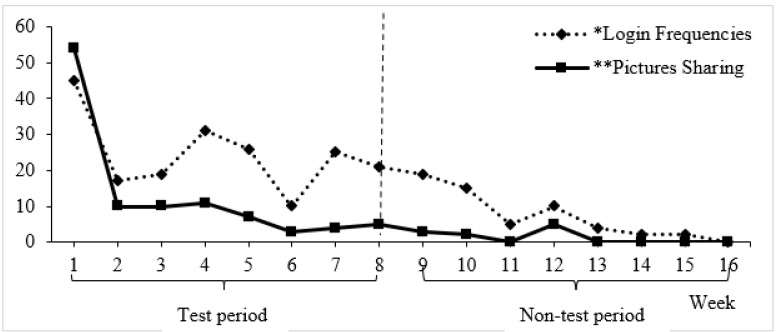
Frequency of logins to the app and sharing of paintings during and after the 8-week intervention.

**Figure 3 healthcare-10-00870-f003:**
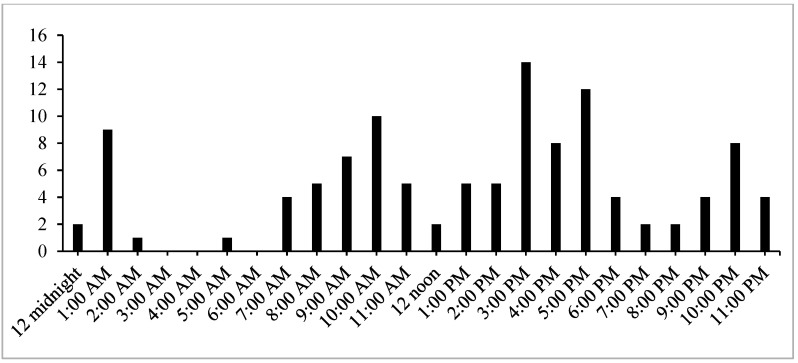
Participants’ usage and time points of using the e-painting app.

**Table 1 healthcare-10-00870-t001:** Characteristics of the participants in the 8-week intervention.

Sociodemographic Characteristics		Frequency	%
Gender	Male	8	28.6
	Female	20	71.4
Age	Below 60	11	39.3
	60 or above	17	60.7
Marital Status	Single	2	7.1
	Married	26	92.9
Places where Educated	Hong Kong	25	89.3
	China or Others	3	10.7
Highest Academic Qualification	Primary or below	4	14.3
	Secondary	14	50.0
	Bachelor’s Degree or above	10	35.7
Employment Status	Retired/Unemployed	22	78.6
	Employed Full-time	4	14.3
	Employed Part-time	2	7.1
Occupation	Managers and Administrators	3	10.7
	Professionals #	8	28.6
	Non-professionals #	9	32.1
	Unclassifiable/Others	8	28.6
Monthly Income(in Hong Kong dollars)	<$2000	12	42.9
	$2001–12,000	5	17.8
	≥$12,001	11	39.3
Number of years spent providing care to persons with dementia	<3 years	12	42.9
	3 to 5 years	5	17.9
	>5 to 10 years	8	28.6
	>10 years	3	10.6

Note. # Professionals refer to doctors, engineers, accountants, teachers, social workers, nurses; Non-professionals refer to clerical support workers, secretaries, typists, cashiers, receptionists, salespeople, and service workers.

**Table 2 healthcare-10-00870-t002:** Frequency of logins and sharing of paintings.

Times	*n* (%)
Frequency of logins	
1 to 8	18 (64.3%)
9 to 16	8 (28.6%)
>16	2 (7.1%)
Frequency of sharing of paintings	
0	8 (28.6%)
1 to 8	16 (57.1%)
9 to 16	3 (10.7%)
>16	1 (3.6%)

**Table 3 healthcare-10-00870-t003:** Psychosocial Assessment before and after the use of the e-painting app.

Psychosocial Well-Being	Pre		Post		Mean Difference	*t*	*p*	Cohen’s D
	Mean	SD	Mean	SD	(SD)			
CZBI-Short	30.71	7.15	33.57	6.89	2.86 (4.87)	3.10	0.004	0.41
SRH	3.14	0.93	3.18	0.77	0.04 (0.92)	0.21	0.839	0.05
PHQ-9	15.25	4.95	15.61	5.03	0.36 (3.20)	0.59	0.560	0.07
mMOS-SS	20.14	6.00	21.32	5.48	1.18 (5.31)	1.17	0.251	0.21

Notes. SD: standard deviation. CZBI-Short measures the caregiving burden, SRH measures self-rated health, PHQ-9 measures depressive symptoms, mMOS-SS measures social support.

## Data Availability

Data are not available to the public.

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
