# Peer review of "The Use of an Electronic Painting Platform by Family Caregivers of Persons with Dementia: A Feasibility and Acceptability Study"

_healthcare, 2022, doi:10.3390/healthcare10050870_

Round 1

Reviewer 1 Report

Dear Authors,

the text is interesting and the application proposal is surely awaited by those concerned with providing care for elderly people with dementia. Applications that help people with dementia in a still low degree to slow the progression of the disease can also be expected further. We tried to work on such solutions in our team. Nevertheless, the way your research and results are presented is too bold for the actual results. After all, what happened? You developed the app, tested it on a rather small group of caregivers (oddly enough it took so many organizations to collaborate on recruiting, line 435-439) and they found the app helpful in relieving stress, but it didn't change their health for the better. Perhaps because they themselves were burdened with diseases additional to depression resulting from the constant care of a sick loved one. Or maybe 8 weeks is too short time to notice an improvement, or maybe the application is not suitable for all applicants, or for several other reasons. However, these doubts did not appear in the text. The assessment is simply positive: it works and the users are satisfied. It is a pity for the research prepared in such a way and published in various international forums for such a fragile conclusion.

The article is written with an anointing worth exploring. I suggest expressing more doubts in the text at every stage of the study, starting with the group of participants (suddenly info that 50? It's hard to count these numbers ... line 229 and whole chapter). I wonder why the study lasted 8 weeks, I wonder why after this time the frequency of entering the application was minimal, if (according to the study) everyone (except for two people) was satisfied and assessed the application as helpful. Helpful but? What?

Why such a long description of the socio-demographic characteristics of the study participants if this information was then not used for any correlation?

Also:

It looks like this information is invalid (line 88-90) and not active:

The trial has been registered in trial register ClinicalTrials.gov (https://clinicaltri- 88 als.gov/ct2/show/NCT02425527). (Archived by WebCite at http://www.webcita- 89 tion.org/6esK11uDH)

Probably the right one is https://clinicaltrials.gov/ct2/show/NCT03850613

Line 21 Result. Recruitment rate was 87.5%.  - not clear

In my opinion, the text should be improved, but undoubtedly worth publishing afterwards.

Reviewer 2 Report

This is an interesting paper, which however requires considerable improvements. 

My detailed comments are as follows:

  1. Introduction, please explain better why is the study undertaken.
  2. The study has 3 objectives (see lines 81-85). Why objective 3 is not disclosed in the abstract? Was the objective 3 of the study achieved? I would expect to see comments on that in discussion section and  in conclusions.
  3. Discussion section requires significant improvement, please increase the number of references over there. The results obtained in the study should be compared to other results provided in the literature so far. Explain how the findings of your study add to the current literature. 
  4. Please add the sample size to the limitations of the study, i.e. although representative, the sample is relatively small.
  5. In Conclusions section, please:
    1. Avoid abbreviations
    2. Provide insightful conclusions, not a short summary.
    3. Provide, very briefly, potential benefits from obtained results in this study

Round 2

Reviewer 1 Report

Dear Authors,

I accept the corrected version of the article.

Author Response

Professional English editing was carried out.

Reviewer 2 Report

The authors improved the paper.  I accept the paper in current form.

I wish authors good luck with further research.

Author Response

Professional English editing was carried out. Thank you for reviewing the paper